# Variation in Bird Eggs—Does Female Factor, Season, and Laying Order Impact the Egg Size, Pigmentation, and Eggshell Thickness of the Eggs of Capercaillie?

**DOI:** 10.3390/ani11123454

**Published:** 2021-12-04

**Authors:** Joanna Rosenbeger, Kamil Pytlak, Ewa Łukaszewicz, Artur Kowalczyk

**Affiliations:** Division of Poultry Breeding, Institute of Animal Breeding, Wroclaw University of Environmental and Life Sciences, 50-375 Wrocław, Poland; 115151@student.upwr.edu.pl (K.P.); ewa.lukaszewicz@upwr.edu.pl (E.Ł.); artur.kowalczyk@upwr.edu.pl (A.K.)

**Keywords:** capercaillie, eggs, eggshell, laying order, pigmentation, intra-species variation

## Abstract

**Simple Summary:**

Birds eggs are unique in the animal kingdom thanks to their different shapes, colours, sizes, and maculation patterns. Generations of people have been fascinated by their variety, thus, egg collecting by scientists and collectors has, in the past, been quite a popular pursuit. Nowadays, this activity is illegal in many jurisdictions, but egg variation has not lost its fascination. Despite extensive research, scientists are yet to determine why birds eggs are so varied, not only between species, but in one species. There are many possible sources of intraspecies egg diversity, such as female factor, laying order, season, and many others. In the presented work, we investigate egg variation and its sources for Capercaillie. We found that size, shape, and pigmentation were not connected to laying order, nor season, but egg traits were highly consistent for individual females. This conclusion indicates that, in the case of Capercaillie, visual identification can be useful in identifying the eggs of different females.

**Abstract:**

Despite numerous studies, intra-species variation in bird eggs is still not well explained. In the presented studies, we investigated the possible sources of this variation: female factor, laying order, and season, using the following traits of Capercaillie eggs as an example: egg size and shape, eggshell lightness, and thickness. Samples were collected for three years from three Capercaillie breeding centres located in different parts of Poland, where birds are kept in conditions close to their natural habitat and have a similar diet. The obtained results showed no significant impact of laying order on egg size, shape, pigmentation, nor eggshell thickness. This indicates that the provided nutrition ensures an adequate supply of minerals for the entire laying period. Most results did not show statistically significant differences between eggs from different breeding centres, but in one breeding centre, eggshells had lighter pigmentation. We assume the observed differences may result from females’ individual features or local environmental conditions. Egg traits were highly consistent for individual females, proving that visual identification can be useful in identifying the eggs of different females.

## 1. Introduction

There is a great variation among bird eggs regarding both morphological features, such as size, shape, or pigmentation [1,2], and the share of individual parts of the egg (shell, yolk, albumen), and their chemical composition [3,4,5]. Particularly diverse are the eggshells, which are most exposed to the external factors, and breeding success largely depends on it [2,6]. The shell, as well as the other products of the evolutionary process, is subjected to selection pressure, manifesting an evolutionary compromise between the expenditure incurred by the female on its production and the provision of adequate resources and conditions to protect the developing embryo [7]. For example, Common cuckoo (*Cuculus canorus*) eggshells are relatively rounder and thicker compared to bird size [8] and they also have a different microstructure [9,10]. This is an evolutionary adaptation to prevent the egg from cracking. In some bird species, pigmentation decreases the risk of predation [11,12,13] or nest parasitism [14,15,16]. Many aspects of eggshells remain unclear; there are many theories about unusual, pyriform egg shape in the Common guillemot (*Uria aalge*) and some of its relatives. This includes reducing the risk of eggs falling from a cliff and lesser contamination by faeces debris [17,18], so researchers are far from finishing their studies.

Given the enormous variety of the Aves class, interspecies egg variation is not a surprising result of the many factors that influence the evolution of each group of birds and particular species. However, the issue of intra-species diversity is much more surprising and complicated. One examples of this is in Yellow-eyed penguins (*Megadyptes antipodes*), in which a positive correlation between female age and eggshell thickness has been observed [19]. Interspecies variation has also been described in Black-legged kittiwake (*Rissa tridactyla*) eggs: they become shorter and broader in older females and differ depending on clutch size and laying sequence. They also tend to be larger towards the north, probably because birds also become larger towards the north of their range [20]. Similar results have been found in the Northern lapwing (*Vanellus vanellus*), in which egg breadth, length, volume, and shape index varies depending on longitude and latitude [21]. Variation in eggshell traits, including eggshell, has also been observed in 15 examined populations of Pied flycatchers (*Ficedula hypoleuca*). However, authors described only little environmental dependency: eggshells were thicker in populations that experienced higher ambient temperatures during egg-laying [22]. Avilés et al. [23] indicated that the eggs of the Eurasian reed warbler (*Acrocephalus scirpaceus*) were greener or bluer in years with lower average air temperature, while in the case of higher than average precipitation, the eggs were shinier. In the case of the Common cuckoo (*Cuculus canorus*), during the years of intense rainfall, the eggs were greener and bluer and they mimicked the host’s eggs better. Geographical and weather variations are not always the only source of variation. Village weaver (*Ploceus cucullatus*) eggs, a species whose nests are parasitized by the Diederik cuckoo (*Chrysococcyx caprius*), have evolved high variation in egg colour between individuals and also show population variation. However, this depends on whether Cuckoos are absent or not [24]. While variation in Weaver eggshells has been described in the context of female identity, in the Russet sparrow (*Passer cinnamomeus*) it was proven that laying order has an important effect on eggshell pigmentation—their last laid eggs were significantly lighter [25,26]. Wendeln’s [27] research indicated that pigmentation may reflect female health and condition in Blue tits (*Cyanistes caeruleus*). In the Common tern (*Sterna hirundo*), it was also revealed that female condition, understood as the body mass before the onset of laying period, was associated with a high egg mass. The intraspecies egg variation was found to be likewise in domesticated birds (Zatorska goose breed), indicating at least a partial genetic source of variation [28].

In one of our previous studies, we indicated that Western capercaillie (*Tetrao urogallus*) eggs laid at the end of the laying season were rounder compared to the beginning of the laying period. In general, in Capercaillie eggs, we also observed large variation in shell pigmentation [29]. However, due to the limited number of samples, a more precise indication of what could affect the noted variation was difficult. This study aimed to expand on the knowledge of present studies and search for sources of variation: what factors exist, and to what extent these factors impact the morphological features of Capercaillie eggshells.

Many publications do not provide complete information and collecting sufficiently extensive material to compare interspecies eggshell variation is often troublesome. Even performing basic measurements may be useful to develop in the future models to determine the factors and to what extent these factors affect egg features. The results obtained from Capercaillie would be especially useful in determining how those patterns are represented in other bird species and birds representing various breeding strategies. Capercaillie, a species endangered in many European countries, seems to be a valuable and interesting research model. Our hypothesis was that egg characteristics, as eggshell pigmentation, would be constant in one year and from season to season, but dimensions may be variable. We expected that the most intra-individual and inter-season variation would appear in eggshell thickness. The aim of the present study was to investigate how and in what way factors such as female identity and laying order impact egg size, pigmentation and eggshell thickness. 

## 2. Materials and Methods

### 2.1. Flock Maintenance Conditions

In all breeding centres, birds are kept in similar conditions, away from human settlement and with limited contact with humans. Flocks were kept throughout the year in wooden roofed aviaries, equipped with a perch, pines (*Pinus sylvestris*) or spruces (*Picea abies*) replaced by fresh material after drying. During the breeding season, all females had access to outdoor aviaries with a variety of natural vegetation and invertebrates where they were able to set up their nests. Once the laying period started, males were removed from the wooden aviaries in order to not disturb the females.

All food and water were provided by the same keepers the birds were accustomed to. To reduce the risk of disease, food and water vessels were washed daily and sterilized. Birds were fed once a day with poultry mash, live crickets, fruits (blueberries, cranberries) and fresh buds of deciduous trees. Needles from pine and spruce trees placed inside aviaries were an additional source of food. Additionally, females were able to search for food in the outside aviaries. Water was acidified with lemon juice (to prevent excessive development of gastrointestinal microbiota in the cecum, and as a consequence, inflammation) and was changed daily. Before and during the laying period, diet was supplemented ad libitum with pigeon grit, rich in crushed shellfish shells.

### 2.2. Egg Evaluation

Unhatched eggs and post-hatched eggshells were collected for three years (2018–2020) from Capercaillie breeding centres in three locations: Wisła Forestry (WF) 49°32″ N, 18°55″ E, Leżajsk Forestry (LF) 50°14″ N, 22°18″ E located in southern Poland, and Głęboki Bród Forestry (GB) 53°59″ N, 23°15″ E located in northern Poland. All eggs were individually numbered and the date of laying, the nest where the egg was laid, and the mother’s identity when known, were written down by bird keepers. Eggs with uncertain female origin (i.e., eggs laid in the nest by other female, eggs laid outside the nest), abnormal eggs (size, pigmentation, or shape, without shells) were excluded from further analyses.

All unhatched eggs were measured (maximum length and width to the nearest 0.01 mm) with the use of electronic callipers. The egg shape index, i.e., the ratio of long to short axis, was calculated. Then, the strength [kg] of eggs with undamaged shells were tested using the EGG Force Reader (ORKA Food Technology Ltd., Ramat HaSharon, Israel). The eggs were placed in the egg cradle horizontally. A force gauge was applied to the upper surface, while the pressure was gradually increased and the moment when the eggshell cracked was recorded to the nearest 0.001 kg. Eggshell pigmentation intensity (lightness, L*) was assessed with a portable colorimeter NH310 (3nh, Shenzhen, China) and was shown in CIE 1976 L*a*b*: colour space. For each eggshell average pigmentation was calculated on three measures on a surface of 8 mm each. Later, the unhatched eggs were opened to determine their status—whether they contained dead embryos and at what age. Eggs containing embryos older than 4 days were excluded from further analysis due to their impact on eggshell thickness [29]. In the case of post-hatched eggshells, durability tests and egg measures were not able to be conducted, so only lightness was measured. All eggshells, both unhatched and post-hatched, were cleaned under running water. Eggshell membranes were mechanically removed. Shortly after drying, the eggshell thickness was measured. Eggshell thickness was measured to the nearest 0.001 mm using a micrometer (Insize3580-25A) with a 0.2 mm spline diameter. Measurements were made of every egg: from the equator, 1 cm from the sharp end, and 1 cm from the blunt end. Each place was measured three times, an average eggshell thickness was calculated from all nine measurements.

### 2.3. Samples Selection

From 51 females, 312 eggshells were obtained in total. This includes 171 post-hatched eggshells and 141 eggs infertile or containing embryos up to the 4th day of development. From WF, 146 samples were obtained: 59 unhatched eggs and 87 post-hatched eggshells (from 28 females); LF 142 samples were collected including 61 unhatched eggs and 74 post-hatched eggshells (from 19 females), and from GB 14 unhatched eggs and 10 post-hatched eggshells (from 4 females). All the samples were used to compare eggshell traits between breeding centres. 

Each planned statistical analysis needed careful sample selection. The eggshell thickness of unhatched eggs and post-hatched eggshells were compared separately to each other (i.e., eggshells thickness comparisons for different breeding seasons were only conducted when at least two different seasons of eggs with the same status were obtained). To analyse the effect of laying order on egg traits, at least three eggs with the same status (no embryo/early dead embryo or post-hatched eggshells) had to be delivered from one female; in total, we analysed 88 eggs from 20 females. To analyse the season effect, a minimum of three eggs with the same status collected from the same female in two subsequent seasons, had to be used. To analyse the similarity of eggs from the same female at least three eggs with the same status had to be compared.

The eggshell thicknesses were separately compared for post-hatched eggshells and unfertilized eggs. Since the embryo effect on eggshell pigmentation is rather unlikely, we compared all eggshells independently from their status.

### 2.4. Statistical Analyze

Statistical analyses were conducted with R statistical package and Statistica (version 8.0, StatSoft, Inc., Kraków, Poland, sp. z o.o.). Firstly, we used the Shapiro–Wilk test to determine whether data had normal distributions. Then, appropriate statistical tests were used: ANOVA (A), *t*-test (T), U-Mann–Whitney test (W), Kruskal–Wallis test (K-W), Pearson (P) or Spearman (S) correlation and coefficient of variation (CV), depending on the specifics of the analysis and data. Abbreviations of the tests used are given for the individual results.

### 2.5. Ethical Note

The National Forestry in Wisła District has the permission (DOP-OZGIZ.6401.03.171.2011.km, dated on: 10 May 2011; expiry date: 31 December 2021) for keeping, reproduction, and collection of the biological materials for experimental purposes of adult and juvenile birds in the Capercaillie Breeding Centre in Wisła Forestry District, Poland. The permission was issued by the General Director of Environmental Protection and signed by Michał Kiełsznia for keeping.

## 3. Results

### 3.1. The Effect of Laying Order on Capercaillie Egg Characteristics

To investigate the influence of the laying order on eggshell thickness, lightness, dimensions, and egg index, analyses were performed using the Pearson or Spearman correlation. In the majority of cases, no statistically significant correlation was found between the laying order and the thickness of the shell near the sharp end of the egg, the blunt end of the egg, or at the equator of the egg. In only two out of 18 females, we found a change of eggshell thickness at the blunt end of the egg. Likewise, the thickness at the equator of the egg changed only in two of the 18 females and in one case, differed near the sharp end of the egg.

The lightness of the eggshell for only two of the 17 tested females changed throughout the breeding season. The egg index correlated with the laying order changed only for two females. Only two out of 19 females had their egg length changed depending on the laying order factor. No changes in egg width or shell durability were found. All detailed calculated correlations are shown in Appendix A.

### 3.2. Variation of the Egg Characteristics in Subsequent Seasons

Analyses were performed using the Kruslall–Wallis test or *t*-test. There was no individual variability of the lightness of the eggshells in subsequent seasons (Table 1). In the case of the eggshell thickness in post-hatched eggshells, only in the case of one female, the eggshell thickness at the sharp end was ambiguous (at the significance level *p* = 0.041). All detailed calculated correlations are shown in Appendix A.

### 3.3. Eggshells Characteristics of Individual Females in Subsequent Seasons

When there were at least three eggs from a given female, we tested the coefficient of variation (CV). CV of eggshell lightness (unfertile eggs/eggs containing early dead embryos and post-hatched eggshells) varied between 0.001–0.114, on average 0.036. CV of egg length varied 0.009–0.154, on average 0.041, egg width 0.001–0.065, on average 0.014, egg shape varied 0.007–0.093, on average 0.039 and eggshell durability 0.000–0.543, on average 0.169 (unfertile eggs). Eggshell thickness varied between 0.023–0.057, on average 0.036 at the blunt end of the egg, 0.007–0.160, on average 0.050 at the sharp end of the egg, 0.007–0.158, on average 0.032 at the equator and 0.012–0.063, on average 0.033 mean eggshell thickness (post-hatched eggshells). In the case of eggs that were unfertilized, eggshell thickness varied between 0.023–0.139, on average 0.059 at the blunt end of the egg, 0.017–0.160, on average 0.075 at the sharp end of the egg, 0.006–0.180, on average 0.054 at the equator and 0.009–0.106, on average 0.053 mean eggshell thickness. Graphs show detailed CV of individual females on the example breeding seasons 2018 (Figure 1) and seasons 2019 and 2020 (Appendix A).

### 3.4. Eggshell Characteristics from Different Breeding Centers

Analyses were performed using the using Kruskall–Wallis test, *t*-test, ANOVA, or U-Mann–Whitney. We did not find any differences in eggshell thickness between breeding centres. Additionally, egg shape index, length, width nor eggshell durability, was not related to the breeding centre. However, eggshells from LF were notably lighter than WF and GB (Appendix A). Detailed egg measurements obtained from WF and LF are shown in Table 2.

## 4. Discussion

In the presented analyses we tested what factors may affect intra-species egg variation in Capercaillie. Thanks to the fact that all eggs came from breeding centres where birds are kept in conditions close to their natural habitat and had similar diets, we were able to minimise the effects of nutrition year by year. This leaves female individual features, subsequent laying seasons, laying order and breeding centres factors that we can treat as local environmental conditions.

Eggshell shape is consistent within female birds [30,31], but on the other hand, in some species, such as House wren (*Troglodytes aedon*) [32], Laughing gull (*Leucophaeus atricilla*) [33] and Common tern (*Sterna hirundo*) [34] has shown that eggs differ depending on laying order. On the other hand, for closer related species, Red-legged partridge (*Alectoris rufa*), egg size, was highly repeatable within females and between breeding seasons [35]. It is known that smaller chicks that hatch from smaller eggs may have poor survival rates [36,37]. Precocials are birds that invest in raising offspring less than altricials, but invest more on egg production, thus the alignment of the egg’s quality, including size, should be high to ensure better chance of survival for the brood. Similar to the Red-legged partridge, in Capercaillie, we found that in most cases eggs were, in fact, similar through all laying order sequences. We found this surprising, especially that our first research indicated that eggs laid at the end of the breeding season were rounder [29]. Detailed research, however, showed no statistical differences for individual females. Capercaillie laying season, depending on environmental conditions, starts in the middle/end of April and ends in early June. The whole clutch is numbered usually 6–8 eggs that are laid every second day. This makes the laying period 12–16 days [38]. We cannot exclude that at described shape change, environmental conditions, such as temperature, have a greater influence than laying order. Interesting results were indicated by Dolenec [39] that egg dimensions in laying order changed, but only in some breeding seasons. It is worth mentioning, that Capercaillie is a species highly exposed to nest predation [40]. Research indicated that smaller eggs in the clutch may be more frequently taken by predators [41], thus no significant variation in size may help to decrease the risk of predation. It is also possible that in captivity, where nourishment resources are not limited, this may prevent egg changes during laying season. Similar research should be conducted on wild populations.

We also found females lay similar (eggshell thickness, lightness) eggs through subsequent breeding seasons. Each female lays similar eggs in one clutch. The biggest variation was found in eggshell thickness. However, it was expected, as this egg trait depends very much on external factors such as bird diet, physiological limitations, and environmental conditions [42]. Still however, coefficient of variation was only around 15% for one female egg length and around 10% for two females’ eggshell thickness. These results indicate that eggs are similar for females year by year. Our conclusions are consistent with the Christians [43] observations, that egg traits depend mostly on the physiological and genetic characteristics of the female.

Historically, in Capercaillie were featured between thirteen [44] and eight subspecies [45]. According to differences in mitochondrial DNA, only two subspecies should be distinguished: *T. u. major* and *T. u. cantabricus* [46]. This, however, did not exclude variations between populations that are adapted to local conditions. Therefore, individual populations may differ in body size, plumage, behaviour, and the characteristics of the eggs. According to Gotzman et al. [47], the Capercaillie eggshells are relatively thick with a low gloss. The eggs are elongated with varying degrees of pointing at the sharp end. Average dimensions, according to various authors were 58 × 41 mm [42], 56.8 × 40.9 mm [48], 57.2 × 41.7 mm (collection of the USSR Academy of Sciences), 58.4 × 41.2 mm (eggs collected from the Pechora population) [49]. Our results show that measured eggs were a little bit smaller (from LF and WF) than the ones described in the literature. They were more similar to eggs from the collection of the USSR Academy of Sciences. Eggs from different breeding centres did not differ, besides eggshell lightness in LF compared to WF and GB. It is unclear if this variation source is genetic (population) or local environment dependent. The rest of the features, such as size, shape, and thickness, were similar. In the case of the thickness measures, they indicate that nutrition in every breeding centre was optimal. We believe that our results, as an indicator of the correct thickness of the shell, may be valuable in helping to control the nutritional requirements of this species in captivity.

Obtained results indicate that Capercaillie egg differentiation probably has a genetic source, thus egg features sometimes may be helpful to identify particular mothers. This information may be useful, for example, when enormous clutches are found, for example in the wild [50]. In captivity, when more than one female is kept together, egg appearance may be an indicator the nest is used by more than one female. This happens often and may lead to aggression and lowering breeding success [38]. Nests that contain eggs of different appearances should be observed more closely due to the greater probability of abandoning the nest or egg destroying.

## 5. Conclusions

The obtained results showed no significant impact of laying order on egg size, shape, and pigmentation, nor eggshell thickness. Additionally, no significant impact of season on eggshell lightness and thickness was observed. Capercaillie, as an example of a precocial species that invests more in egg quality rather than parental care, should have evolved to maintain the quality of eggs at a constant, high level. In LF breeding centre, eggshells were lighter than WF and GB breeding centres, which may be due to local environmental conditions or female individual features. Measured egg traits were consistent for individual females, particularly year by year, indicating its genetic sources. This makes the visual identification of mothers based on egg appearance possible. As expected, the biggest variation was found for eggshell thickness, probably due to the fact that this trait is depended on external factors: nutrition, physiological limitations, and environmental conditions.

## Figures and Tables

**Figure 1 animals-11-03454-f001:**
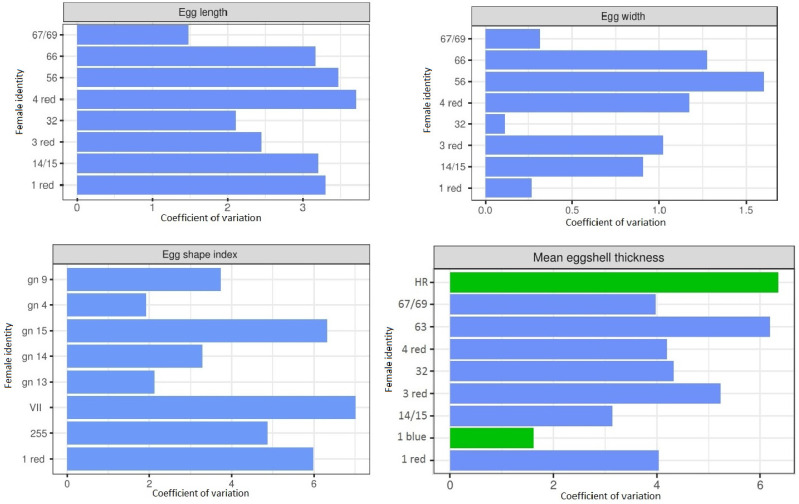
Coefficient of variation for egg length, width, egg shape index and mean eggshell thickness for particular females in the year 2018. The compared unfertilized eggs are marked in blue, the post-hatched eggshells are marked in green.

**Table 1 animals-11-03454-t001:** Variation in eggshells lightness [L*] over the following seasons (2018, 2019, 2020) for the same females. K-W means performed test was Kruskall–Wallis, T performed test was *t*-test. The number of eggs (*n*) was given for each L*.

Female ID	Season/Year			
	L * in 2018	L * in 2019	L * in 2020	*p*-Value
6 red	69.768 (*n* = 4)	67.043 (*n* = 5)	72.1 (*n* = 3)	0.9832 (K-W)
2 green	*	72.944 (*n* = 3)	73.8 (*n* = 7)	0.296 (T)
	*	1.19	2.471	
8 blue	*	69.63 (*n* = 4)	69.735 (*n* = 9)	0.467 (T)
	*	1.758	2.084	
25 green	68.439 (*n* = 5)	*	66.5 (*n* = 3)	0.157 (T)
	2.19	*	2.798	
56	71.062 (*n* = 5)	72.639 (*n* = 7)	*	0.053 (T)
	1.202	1.691	*	
73	*	67.692 (*n* = 3)	67.86 (*n* = 5)	0.449 (T)
	*	2.189	1.434	
60	*	64.31(*n* = 3)	67.701 (*n* = 5)	0.058 (T)
	*	2.796	2.382	
23	74.236 (*n* = 4)	*	70.938 (*n* = 8)	0.07 (T)
	2.579	*	3.631	

*—no samples obtained from female in year.

**Table 2 animals-11-03454-t002:** The length, width, shape, and eggshell lightness of Capercaillie eggs collected from Wisła Forestry (WF) (*n* = 59) and Leżajsk Forestry (LF) (*n* = 61).

Measured Trait	Average	SD	Min	Max
Egg length (mm)	WF 55.421	WF 2.749	WF 49.330	WF 61.260
	LF 56.766	LF 2.892	LF 44.660	LF 62.230
Egg width (mm)	WF 41.286	WF 1.059	WF 38.790	WF 44.300
	LF 41.281	LF 0.960	LF 36.430	LF 43.130
Egg shape	WF 1.342	WF 0.049	WF 1.255	WF 1.444
	LF 1.375	LF 0.068	LF 1.226	LF 1.536
Eggshell lightness	WF 70.364	WF 4.357	WF 57.573	WF 79.760
	LF 70.166	LF 4.302	LF 54.523	LF 78.997

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
