# Peer review of "Variation in Bird Eggs—Does Female Factor, Season, and Laying Order Impact the Egg Size, Pigmentation, and Eggshell Thickness of the Eggs of Capercaillie?"

_animals, 2021, doi:10.3390/ani11123454_

Round 1

Reviewer 1 Report

The authors of the study studied the possible sources of this variation: female factor, laying order, and season, using the following traits of Capercaillie egg as an example: egg size and shape, eggshell lightness, and thickness. They found that differences in egg characteristics may result from females' individual features or local environmental conditions. Egg traits were highly constant for individual females, so that proves that visual identification can be useful in identifying the eggs of the different females.

My comments/suggestions/recommendation are shown herein and detailed in the attached copy.

  1. Editing of language of the Ms are essential?
  2. Plz add the theory/hypothesis of this work in the introduction section? 
  3. How is the age of embryonic mortality was measured, pLz declare with reference?
  4.  In all tables Plz provide the number of eggs used.
  5. L 267, Plz use mineral nutrition as most fit for eggshell thickness.
  6.  The conclusion section is absent, Plz complete?
  7. Plz update reference up to 2021

Author Response

Dear reviewer, thank you for all of your comments and advises. They were really helpful to improve our manuscript. We tried to answer them as best as possible. We hope, you will be satisfied.

Below we give detailed responses to your comments.

  1. Plz add the theory/hypothesis of this work in the introduction section? - Done
  2. How is the age of embryonic mortality was measured, pLz declare with reference? When determining the day of embryo development, we relied on our previous experiences. Due to fact that we exclude embryos older than 4 days, we did not need any tables describing the embryo development. We exclude everything that was bigger than pea size. They were only 4 eggs containing dead embryos, including one 4 days old. 
  3.  In all tables Plz provide the number of eggs used. - Done
  4. L 267, Plz use mineral nutrition as most fit for eggshell thickness.- We added section about bird maintenance, including nutrition
  5.  The conclusion section is absent, Plz complete? Done. It was our mistake to upload wrong file.
  6. Plz update reference up to 2021 – We expanded discussion

Reviewer 2 Report

Dear authors,

First at all, I want to thank for the possibility to review this manuscript. The paper addresses well-studied issues, but some of the issues are described very briefly. I suggest you make corrections before proceeding further with the paper:

Line 2: The use of the name of the studied species in the title could be useful for researchers that are looking for information on this species. So, I suggest replacing “birds” with “Capercaillie” in the title.

Line 9: “sizes, and maculation patterns”.

Line 33: At the end of the introduction, it would be necessary to add a paragraph indicating which are the main objectives studied in this research. Moreover, it would be advisable to add information about the status of the Capercaillie, in order to give an idea of the importance of this study.

Line 96: A subsection is missing indicating the conditions in which the animals were found and the diets of these.

Lines 109-110: Please indicate the units in which the eggshell’s strength was measured in the eggs.

Lines 113-114: You indicate that you used the CIELab color space. Strictly, the technical name of this test would be CIE 1976 L*a*b*:

Moreover, in line 113 you indicate that you assessed only lightness. Using the CIE 1976 L*a*b* method, you measure the lightness, red/green, and yellow/blue coordinates. Please clarify it. If you have not used the red/green and yellow/blue coordinates, explain why. These coordinates could give valuable information about the evolution of Capercaillie eggs.

Line 132: GB or GBF? Please be uniform throughout the manuscript.

Line 148: Please develop how the choice of one statistical method or another was made. The information on the statistical analyzes is limited and the results could not be reproduced or discussed by not indicating which analysis is carried out in each situation.

Line 154: 2.4. point instead of 2.5.

Line 158: Please delete the point after “Capercaillie”.

Line 162: The first sentence of each sub-point in the results section corresponds to the M&M section.

Line 164: Please specify what characteristics you mean when you talk about “various characteristics”.

Line 173: “Shape index” is the correct denomination.

Line 185 – Table 1: I do not understand this table. What results in columns 2, 3 and 4 (years) correspond to the first column (Female id)? Please clarify it.

Line 218: Discussion lacks a scientific depth explaining why each of the characteristics of the egg. The fact of investigating why each of the results occurs would contribute a higher quality to the manuscript, since the novelty of the results obtained does not arouse great scientific interest (you have cited a work of yours in the same species). Therefore, I think it would be convenient to carry out a greater bibliographic review on the subject (there are many articles on this subject, there are even exclusive ornithology journals) and carry out a more in-depth discussion.

Line 222: You do not explain the diets they provide in each breeding center, so you can not discuss the effects of nutrition.

Lines 271-272: Using the appearance of the egg as an indicator of the egg id is a very archaic and unscientific method. Ideally, with the results that have been obtained, develop a statistical tool that allows, with a series of egg measurements, to assign this egg to a specific female. Furthermore, this tool must be statistically tested and validated. Could you apply it in this study?

Line 275: Where are the conclusions?

Author Response

Dear reviewer, thank you for all of your comments and advises. They were really helpful to improve our manuscript. We tried to answer them as best as possible. We hope, you will be satisfied.

Below we give detailed responses to your comments.

Line 2: The use of the name of the studied species in the title could be useful for researchers that are looking for information on this species. So, I suggest replacing “birds” with “Capercaillie” in the title. -  we added „on example of the Capercaillie” in title. We agree but also believe that sometimes researchers do not have precised species to look at, but rather general phenomena.

Line 9: “sizes, and maculation patterns”. - corrected

Line 33: At the end of the introduction, it would be necessary to add a paragraph indicating which are the main objectives studied in this research. Moreover, it would be advisable to add information about the status of the Capercaillie, in order to give an idea of the importance of this study. - as suggested, we have added this

Line 96: A subsection is missing indicating the conditions in which the animals were found and the diets of these. - done

Lines 109-110: Please indicate the units in which the eggshell’s strength was measured in the eggs. - done

Lines 113-114: You indicate that you used the CIELab color space. Strictly, the technical name of this test would be CIE 1976 L*a*b*:

Moreover, in line 113 you indicate that you assessed only lightness. Using the CIE 1976 L*a*b* method, you measure the lightness, red/green, and yellow/blue coordinates. Please clarify it. If you have not used the red/green and yellow/blue coordinates, explain why. These coordinates could give valuable information about the evolution of Capercaillie eggs. – it is our mistake. In our previous studies we measured only lightness and this time we made the same as a factor that is the most intuitive. Next time we would include red/green, and yellow/blue coordinates as you suggest.

Line 132: GB or GBF? Please be uniform throughout the manuscript. - done

Line 148: Please develop how the choice of one statistical method or another was made. The information on the statistical analyzes is limited and the results could not be reproduced or discussed by not indicating which analysis is carried out in each situation. We added some information to this section, however as we used many tests, depending data specificity, we decided it will be more clear to provide them with specific results, rather in M&M. So the reader would not go up and down to know in which case which test was used. We also used abbreviations in tables to make it more clear. I think that deviation from the scheme/rules allows for better ordering of information.

Line 154: 2.4. point instead of 2.5. - numbering, including the addition of sections „Flock maintenance conditions” corrected

Line 158: Please delete the point after “Capercaillie”. - done

Line 162: The first sentence of each sub-point in the results section corresponds to the M&M section. It is true, but I decided it will be more clear for readers to start each sub-point with this information rather give all those information above. We used many tests and describing all of this in M&M would get the reader to go back to this section to see what analysis and what test was used for. I think that deviation from the scheme/rules allows for better ordering of information.

Line 164: Please specify what characteristics you mean when you talk about “various characteristics”. -done

Line 173: “Shape index” is the correct denomination. - done

Line 185 – Table 1: I do not understand this table. What results in columns 2, 3 and 4 (years) correspond to the first column (Female id)? Please clarify it. In first column is female identity (to prevent mistakes when analyzing data we called them according to color rings they had; breeding centers had different ways to ring them: WF had the same ring colors but different number, while in LF they might be for example  more than one females with ring number 6 but different colors). In columns 2-4 are presented lightness for subsequent seasons and below them value of the test used. I tries to make it more clear in table.

Line 218: Discussion lacks a scientific depth explaining why each of the characteristics of the egg. The fact of investigating why each of the results occurs would contribute a higher quality to the manuscript, since the novelty of the results obtained does not arouse great scientific interest (you have cited a work of yours in the same species). Therefore, I think it would be convenient to carry out a greater bibliographic review on the subject (there are many articles on this subject, there are even exclusive ornithology journals) and carry out a more in-depth discussion. – As you suggested, we expanded discussion

Line 222: You do not explain the diets they provide in each breeding center, so you can not discuss the effects of nutrition. – I added sections in material and methods. In general, in all breeding centres birds have very similar nutrition. They exchange their observation and experience.

Lines 271-272: Using the appearance of the egg as an indicator of the egg id is a very archaic and unscientific method. Ideally, with the results that have been obtained, develop a statistical tool that allows, with a series of egg measurements, to assign this egg to a specific female. Furthermore, this tool must be statistically tested and validated. Could you apply it in this study? I think much more eggs from the same females should be gathered. This however is troublesome because we need to have unhatched eggs to compare. Unfortunately we are not allowed to interfere when females incubate. It is great idea, but we need many years to collect big enough data base.

Line 275: Where are the conclusions? – Done. It was our mistake to upload wrong file.

Round 2

Reviewer 2 Report

Dear authors,

The manuscript has been substantially improved and the added information has increased the quality of the study.

However, in the conclusion, you only present the results but the true conclusions that have been obtained in the discussion are not synthesized.

I think the conclusion should be rewritten and modified, before the publication of this article in Animals.

Author Response

Dear reviewer,

Once again, thank you for your comments. They were really valuable and helped us to improve the manuscript.  

According to your just suggestion, we improved final conclusions, adding most important thoughts from discussion.

Yours sincerely

Joanna Rosenberger
